# Fe$_3$O$_4$ Nanoparticles Loaded Bentonite/Sawdust Interface for the Removal of Methylene Blue: Insights into Adsorption Performance and Mechanism via Experiments and Theoretical Calculations

**Mohamed A. Barakat** [1,2] , **Rajeev Kumar** [1] , **Riyadh F. Halawani** [1] , **Bandar A. Al-Mur** [1] **and Moaaz K. Seliem** [3,*]

1 Department of Environmental Sciences, Faculty of Meteorology, Environment and Arid Land Agriculture, King Abdulaziz University, Jeddah 21589, Saudi Arabia
2 Central Metallurgical R & D Institute, Helwan 11421, Egypt
3 Faculty of Earth Science, Beni-Suef University, Beni-Suef 62511, Egypt
* Correspondence: debakyms@yahoo.com

**Abstract:** Herein, magnetite nanoparticles (MNPs) synthesized from altered basalt were used to develop a composite with H$_2$O$_2$–activated bentonite (BE) and fibrous sawdust (SD). The as-prepared BE/SD–MNPs were characterized by FTIR, FESEM, TEM, TGA, DSC, and Zeta potential techniques and utilized as an effective multifunctional composite for removing methylene blue (MB). The adsorption isotherms of MB at 25–55 °C were analyzed via kinetics, classical, and advanced statistical physics models. Theoretically, the pseudo-second-order of kinetics and the Freundlich isotherms model fit the experimental data well without microscopically clarifying the adsorption mechanism. Studying a multilayer model's steric and energetic parameters was a reliable approach to understanding the MB uptake mechanism at the molecular scale. Sterically, the removed MB molecules offered a combination of horizontal and vertical geometry (i.e., mixed orientation). The MB adsorption capacity at saturation ($Q_{sat}$) increased from 829 to 849 mg/g with temperature, suggesting endothermic interactions. Energetically, the MB uptake by BE/SD–MNPs was controlled by physical interactions (i.e., adsorption energy < 20 kJ/mol). The BE/SD–MNPs retained more than 85% of the MB uptake after five adsorption-desorption rounds. Overall, this study aimed to understand the MB adsorption mechanism using a magnetic clays/lignocellulosic interface such as the utilized BE/SD–MNPs composite as a promising strategy in wastewater remediation.

**Keywords:** bentonite; sawdust; magnetic nanoparticles; methylene blue; kinetics; equilibrium; thermodynamics; statistical models

## 1. Introduction

Pollution of water bodies through the effluents of synthetic dyes is a severe environmental problem due to the high toxicity and stability of these organic contaminates [1–3]. Every year, more than $7 \times 10^5$ tons of dyes are produced worldwide, and nearly 10–15% of these water-coloring agents are released into wastewater during their production and dyeing process [4]. Numerous industries, such as food, textiles, and cosmetics, use dyes as coloring agents to obtain products having distinguishing colors [5]. In particular, methylene blue (MB) is recognized as a stable cationic dye under an extensive range of heat, light, and chemical agents [4,6,7]. MB was selected in the current study due to its high stability and solubility (40 g/L), and the contribution of this dye in different industrial applications [8]. Therefore, releasing MB into water bodies without any preliminary treatment causes several environmental and human health risks.

Consequently, finding effective decontamination techniques to purify wastewater containing MB is mandatory. Many methods, including adsorption, ultrafiltration, advanced

oxidation, coagulation, membrane technologies, or biological treatment were utilized to capture organic dyes from solutions [9–11]. Among all treatment approaches, an adsorption system is desired in water purification due to its versatility, non-toxicity, high performance, and low cost [12–14]. Nevertheless, the high cost of some adsorbents usually prevents their broad uses and, therefore, utilizes low-cost adsorbents with a significant number of active sites in water remediation.

Bentonite (BE) is one of the most studied clays utilized in water purification due to its high natural accessibility, chemical structure, great surface area, high ion exchange capacity, high performance, and eco-friendly characteristics [15,16]. The mineral composition (mainly Na-montmorillonite) and structure of the BE facilitate its modification via varied chemical and physical treatments generating materials with new active adsorption sites for water contaminants [16]. For instance, using hydrogen peroxide ($H_2O_2$) in the alteration of clay minerals resulted in a suitable approach by enabling the addition of different modifiers in their structures, producing superior adsorbents for water pollutants [17,18]. Wood sawdust (SD), a common lignocellulosic waste, is recognized as low-cost adsorbent for the uptake of toxic dyes from wastewater [19,20]. Nevertheless, removing SD from aqueous solutions after adsorption is very hard and a long period of time is required to achieve the SD-liquid separation [19]. To easily separate SD from solutions after the adsorption technique, magnetic $Fe_3O_4$ as supported nanoparticles were used [19]. Magnetic $Fe_3O_4$ nanoparticles (MNPs) displayed positive results as cationic and anionic dyes adsorbents [18,21]. Decoration of adsorbates by MNPs resulted in the addition of new functional groups to interact with organic dyes and, thus, the adsorbent efficiency significantly increased [21]. Furthermore, magnetic adsorbents can be removed easily from the treated solutions by a magnet (i.e., a simple and low-cost method) [18].

Clays/lignocellulosic biomasses materials supported by iron oxide nanoparticles are reliable for producing promising multifunctional composites with higher adsorption capacities. Thus, the novelty of this study is to fabricate a multifunctional composite from bentonite/sawdust supported by magnetic $Fe_3O_4$ nanoparticles (BE/SD–MNPs) as an effective strategy to enhance the decontamination of MB dye from water. The classical isotherm models (e.g., Langmuir and Freundlich) and steric and energetic parameters derived from the advanced statistical physics models (e.g., monolayer, double layer, and multilayer) are also utilized. Interpretation of these physicochemical parameters [i.e., the number of MB molecules removed per active sites of the BE/SD–MNPs ($n$), the density of BE/SD–MNPs receptor sites ($D_M$), the formed number of MB layers ($1 + N_2$), the adsorption capacity at saturation ($Q_{sat}$), and the adsorption energy ($\Delta E$)] at different temperatures can describe macroscopically and microscopically the geometry and interactions mechanism between the BE/SD–MNPs adsorbent and MB dye. Results presented in this study offer a distinct approach to increasing the value added of accessible and low-priced materials for water purification. Furthermore, deep insights into hazardous MB adsorption mechanisms using the recyclable and multifunctional BE/SD–MNPs adsorbent were also presented.

## 2. Materials and Methods

### 2.1. Materials

The bentonite was obtained from Shanghai Biochemical Co., Ltd., Shanghai, China. Sawdust was locally attained from a sawmill in Beni Suef governorate, Egypt. Weathered basalt heated at 900 °C/3 h was used as Fe(III) source. The chemicals used in this article were: Methylene blue (MB, $C_{16}H_{18}ClN_3S$, $\lambda_{max}$ 664 nm, 319.85 g/mol), hydrogen peroxide (30%, $H_2O_2$), and iron (II) sulfate ($FeSO_4._2O$) were supplied from Loba Chemie, Pvt. Ltd., Mumbai, India. For adjusting the solution pH value, sodium hydroxide (0.1 M NaOH), and hydrochloric acid (0.1 M HCl) were used.

### 2.2. Preparation of BE/SD–MNPs Composite

The collected SD was washed with deionized water to remove unwanted materials. Then, the SD was dried at 65 °C/48 h and ground to pass through a 100 μm mesh sieve.

The bentonite sample was modified by $H_2O_2$ using a previously described method [22] as follows: 3.0 g of the BE was added to a clean beaker with 25 mL of distilled water with stirring for 60 min. Then, 15 mL of $H_2O_2$ was added to this $BE/H_2O$ mixture with continuous stirring for 2 h at 40 °C to produce BE with open pits and holes. A mass of the SD powder (0.75 g) was added to the BE slurry and agitated for 3 h at 50 °C.

Fe$_3$O$_4$ nanoparticles (MNPs) were prepared in another beaker using activated weathered basalt via the chemical precipitation method described in our previous article [5]. Typically, 10 mL of $NH_4OH$ (25%) as a precipitating agent was added slowly to a mixture of 1.5 g of $FeSO_4 \cdot 7H_2O$, 3.0 g of the heated weathered basalt, and 25 mL of distilled water. This black mixture was subjected to mechanical stirring for 90 min. Finally, the BE/SD of the first baker and the fabricated MNPs of the second one were mixed under vigorous agitation for 120 min. The formed magnetic BE/SD–MNPs composite was separated by a magnet, washed with deionized water numerous times, and dried at 70 °C for 24 before characterization and its application for MB adsorption. Figure 1 showed a schematic diagram for the BE/SD–MNPs composite synthesis procedure.

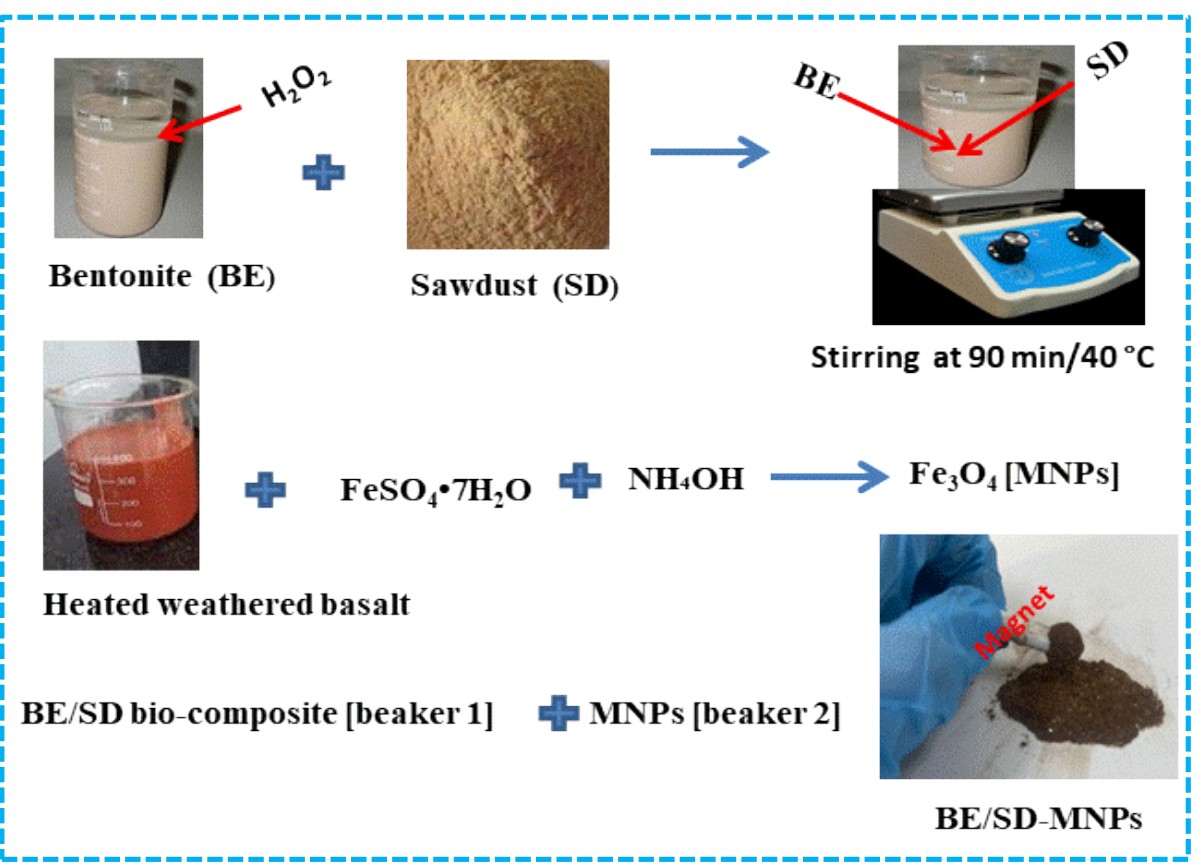

**Figure 1.** The schematic representation of BE/SD–MNPs composite the fabrication.

### 2.3. Characterization of BE/SD–MNPs

The functional groups of the as-synthesized BE/SD–MNPs composite were recognized using Fourier transform infrared (Bruker FTIR-2000 spectroscopy) within a range of 4000–400 cm$^{-1}$. Thermal analyses (e.g., thermogravimetric analysis (TGA) and differential scanning calorimetry (DSC)) of BE/SD–MNPs were performed in the temperature range of 25–1000 °C with a heating rate of 10 °C/min. The Zeta potential of the BE/SD–MNPs composite was measured at 25 °C using Malvern Zetasizer Nano ZS90. The morphological properties of BE/SD–MNPs sample was observed using a Field emission scanning electron microscope (FESEM, Sigma 500 VP, Berlin, Germany) and Transmission electron microscope (TEM, JEOL-JEM2100, Tokyo, Japan).

### 2.4. Kinetics Studies of MB Adsorption onto BE/SD–MNPs

MB kinetics were considered in the adsorption system containing a mixture of 25 mg of BE/SD–MNPs and 25 mL of MB solution with a 150 mg/L concentration. The adsorbate–adsorbent mixtures were mixed at 220 rpm at different interaction times (i.e., 5, 15, 30, 45, 60, 75, 90, 180, and 360 min), and the solution was pH 8.0. The liquid phases were separated by centrifuging, and the removed amount ($q_t$) of MB after each time interval was calculated using the next formula:

$$q_t \text{ (mg/g)} = (C_0\text{–}C_t)\frac{V}{m} \tag{1}$$

where $C_0$ (mg/L) is the initial MB concentration $C_t$ is the rest dye concentration after time ($t$). $V$ is the volume of solution ($L$), and $m$ is the utilized mass of the BE/SD–MNPs (g). To study the adsorption kinetics of MB on the BE/SD–MNPs composite and find the possible rate-controlling step, the pseudo-first-order [23], the pseudo-second-order [24], and intra-particle diffusion [25] equations were used as follows:

$$\ln(q_e - q_t) = \ln q_e - k_1 t \qquad \text{Pseudo} - \text{first} - \text{order} \tag{2}$$

$$t/q_t = (1/k_2 q_e^2) + t/q_e \qquad \text{Pseudo} - \text{second} - \text{order} \tag{3}$$

$$q_t = k_p t^{1/2} + C \qquad \text{Intra} - \text{particle diffusion} \tag{4}$$

where $k_1$ (min$^{-1}$) and $k_2$ (g/mg min) signify the uptake rate constants of the applied first-order and the second-order kinetics, respectively. The $k_p$ (mg/g min) and $C$ (mg/g) characterize the rate constant and the intercept value of the intra-particle model, respectively.

### 2.5. Equilibrium Studies of MB Adsorption onto BE/SD–MNPs

A standard MB solution (1.0 g/L) was diluted to prepare the required concentrations (i.e., 50–400 mg/L) for the isotherm experiments. Equilibrium studies associated with MB uptake by the BE/SD–MNPs were achieved at pH 8.0 and adsorption temperatures of 25, 40, and 55 °C using 25 mg of the BE/SD–MNPs adsorbent. The BE/SD–MNPs and MB mixtures were mixed at 150 rpm/120 min by a SHO–2D digital orbital shaker. The solid-liquid phases were separated by centrifuging, and the MB concentrations in solutions were determined by a double beam UV–visible (Shimadzu UV 1601, Japan) spectrophotometer. The equilibrium adsorption quantities of MB ($q_e$) were determined using the next relationship.

$$q_e \text{ (mg/g)} = (C_0\text{–}C_e)\frac{V}{m} \tag{5}$$

where $C_0$ (mg/L) is the initial concentration of MB, $C_e$ (mg/L) is the remaining concentration of MB dye at equilibrium, $V$ (L) is the solution volume, and $m$ (g) is the mass of the BE/SD–MNPs adsorbent. All MB adsorption experiments were repeated three times, and the mean values of the results have been utilized for data evaluation, with the error always being below ±5%.

### 2.6. Classical Modeling of MB Adsorption onto BE/SD–MNPs

The Langmuir [26] and Freundlich [27] equations fit the MB adsorption on BE/SD–MNPs at equilibrium. The non-linear relations of these classical models are presented in Equations (6) and (7).

$$q_e = \frac{q_{\max} K_L C_e}{(1 + K_L C_e)} \qquad \text{(Langmuir model)} \tag{6}$$

$$q_e = K_F C_e^{1/n} \qquad \text{(Freundlich model)} \tag{7}$$

in which $q_{\max}$ (mg/g) and $K_L$ (L/mg) represent the maximum captured amounts of the adsorbate and the Langmuir constant, respectively. $K_F$((mg/g)(mg /L)$^{-1/n}$)) and $n$ are the normal constants of the Freundlich model related to the adsorption capacities and

intensities, respectively. Equations (8) and (9) were used to find the best-fitted adsorption model according to the determination coefficient ($R^2$) and the Chi-square ($\chi^2$) values [18]:

$$R^2 = 1 - \frac{\sum(q_{e,\text{exp}} - q_{e,\text{cal}})^2}{\sum(q_{e,\text{exp}} - q_{e,\text{mean}})^2} \tag{8}$$

$$\chi^2 = \sum \frac{(q_{e,\text{exp}} - q_{e,\text{cal}})^2}{q_{e,\text{cal}}} \tag{9}$$

where $q_{e,\text{exp}}$ (mg/g) and $q_{e,\text{cal}}$ (mg/g) are the experimental and theoretical values of MB adsorption capacities, respectively.

*2.7. Advanced Modeling of MB Adsorption onto BE/SD–MNPs*

Characteristically, the Freundlich and Langmuir models are insufficient to understand the geometry (e.g., horizontal against vertical) and the uptake mechanism (e.g., multi-docking against multi-interactions) due to the lack of a physical meaning associated with the parameters of these classical models. Thus, advanced statistical physics models (e.g., monolayer, double layer, and multilayer) with different theories were used to consider the interactions between the BE/SD–MNPs and MB molecules at the molecular scale.

2.7.1. Monolayer Model (M1)

The M1 assumes that MB removal by the BE/SD–MNPs was controlled by a monolayer process (i.e., a single removed layer related to the interface between MB molecules and the BE/SD–MNPs surface is offered). Equation (10) provides the mathematical expression of the utilized AMM [5].

$$q_e = \frac{nD_M}{1 + \left(\frac{c_{1/2}}{c}\right)^n} \tag{10}$$

where $C_{1/2}$ displays the concentration at half-saturation associated with the formed MB layer on the BE/SD–MNPs surface.

2.7.2. Double Layer Model (M2)

The M2 suggests that the MB removal process is directed by the formation of two MB layers with two dissimilar adsorption energies (i.e., the principal interface energy ($\Delta E_1$) is named for the monolayer model, while the second one ($\Delta E_2$) characterizes the molecules–molecules interactions of MB dye). The mathematical expression of the M2 is suggested by [17].

$$q_e = nD_M \frac{\left(\frac{c}{c_1}\right)^n + 2\left(\frac{c}{c_2}\right)^{2n}}{1 + \left(\frac{c}{c_1}\right)^n + \left(\frac{c}{c_2}\right)^{2n}} \tag{11}$$

in Equation (11), two concentrations at half-saturation (i.e., $C_1$ and $C_2$) are recognized, which are related to the removed two MB dye layers.

2.7.3. Multilayer Model (M3)

Based on the suggestion of M3, the adsorption system is described by the presence of limited layer number of the investigated MB dye directed by different energies (i.e., $\Delta E_1$ and $\Delta E_2$ [28]. The highest adsorption energy $\Delta E_1$ was attributed to the direct contact between the first removed MB layer (stable number) and BE/SD–MNPs active sites, while the lowest energy $\Delta E_1$ was due to the MB–MB interface [17]. Therefore, the total MB layers number is equivalent to $1 + N_2$ [28], and $N_2 = 0$, 1 or >1, which parallels a monolayer ($N_2 = 0$),

double-layer ($N_2 = 1$), and multilayer process ($N_2 > 1$) [5]. The mathematical expressions used in the calculation the M3 parameters are given in Equations (12)–(17) [5,17,28]:

$$q_e \;=\; n\,D_M \frac{F_1(c) + F_2(c) + F_3(c) + F_4(c)}{G(c)} \tag{12}$$

$$F_1(c) \;=\; -\frac{2\left(\frac{c}{c_1}\right)^{2n}}{1 - \left(\frac{c}{c_1}\right)^n} + \frac{\left(\frac{c}{c_1}\right)^n \left(1 - \left(\frac{c}{c_1}\right)^{2n}\right)}{\left(1 - \left(\frac{c}{c_1}\right)^n\right)^2}, \tag{13}$$

$$F_2(c) \;=\; \frac{2\left(\frac{c}{c_1}\right)^n \left(\frac{c}{c_2}\right)^n \left(1 - \left(\frac{c}{c_2}\right)^{n\,N_2}\right)}{1 - \left(\frac{c}{c_2}\right)^n}, \tag{14}$$

$$F_3(c) \;=\; -N_2 \frac{\left(\frac{c}{c_1}\right)^n \left(\frac{c}{c_2}\right)^n \left(\frac{c}{c_2}\right)^{n\,N_2}}{1 - \left(\frac{c}{c_2}\right)^n}, \tag{15}$$

$$F_4(c) \;=\; \frac{\left(\frac{c}{c_1}\right)^n \left(\frac{c}{c_2}\right)^{2n} \left(1 - \left(\frac{c}{c_2}\right)^{n\,N_2}\right)}{\left(1 - \left(\frac{c}{c_2}\right)^n\right)^2}, \tag{16}$$

$$G(c) \;=\; \frac{\left(1 - \left(\frac{c}{c_1}\right)^{2n}\right)}{1 - \left(\frac{c}{c_1}\right)^n} + \frac{\left(\frac{c}{c_1}\right)^n \left(\frac{c}{c_2}\right)^n \left(1 - \left(\frac{c}{c_2}\right)^{n\,N_2}\right)}{\left(1 - \left(\frac{c}{c_2}\right)^n\right)^2} \tag{17}$$

Generally, many operating states can be related to the steric $n$ and $N_2$ parameters of the M3 as follows [5,28]:

State 1: $n$ and $N_2$ are free adaptable (i.e., multilayer model)
State 2: $n$ is adjustable and $N_2$ = zero (i.e., monolayer model)
State 3: $n$ is variable and $N_2 = 1$ (i.e., double layer model)
State 4: $n$ is inconstant and $N_2 = 2$ (i.e., triple layer model)
State 5: $n = 1.0$ and $N_2$ = zero (i.e., the Langmuir model)

In addition to the $R^2$, the root mean square error (RMSE), calculated as presented in Equation (14), was used to find the greatest advanced model for MB uptake by BE/SD–MNPs composite [5,17].

$$RMSE \;=\; \sqrt{\frac{\sum_{i=1}^{m}\left(Q_{i\,\text{cal}} - Q_{i\,\text{exp}}\right)^2}{m' - p}} \tag{18}$$

where $m'$ signifies the experimental data and $p$ is the number of adaptable parameters.

### 2.8. Regeneration of BE/SD–MNPs Adsorbent

Reutilizing any product as an adsorbent is a significant concern for decreasing costs associated with wastewater purification in industrial systems [17]. The BE/SD–MNPs composite was recycled at room temperature (25 °C) using 100 mL of sodium hydroxide (0.5 M NaOH) as a desorbing element. A rotary shaker agitated the as-synthesized BE/SD–MNPs composite overloaded by MB molecules for 120 min at 200 rpm. This adsorption/desorption method was repeated five times. At the end of each desorption cycle, the BE/SD–MNPs were extensively washed with distilled water and completely dried at 65 °C before the next round.

## 3. Results and Discussion

### 3.1. Characterization of BE/SD–MNPs Composite

FESEM and TEM studies were used to identify the interface between bentonite, sawdust, and $Fe_3O_4$ nanoparticles (i.e., morphology and internal structure of the developed BE/SD–MNPs composite), as shown in Figure 2. The BE sample (Figure 2a) displays stacked layers of montmorillonite agglomerated with each other in masses with smooth surfaces and lenticular secondary irregular pores (i.e., characteristically Wyoming bentonite) [16,29]. The interaction between bentonite and sawdust resulted in the formation of a mixed phase of bentonite sheets and longitudinal particles of SD (i.e., BE/SD), as shown in Figure 2b–d. This BE/SD interaction caused irregular fibrous associated with SD particles with clear interface with montmorillonite sheets. Moreover, BE/SD-supported by $Fe_3O_4$ nanoparticles (Figure 2e–g) displays the decoration of BE/SD by the agglomerated spherical magnetic nanoparticles with dissimilar diameters less than 100 nm forming the BE/SD–MNPs composite. The $Fe_3O_4$ looked as if spherical nanoparticles loaded the BE/SD matrix as separated grains. Furthermore, MNPs filled the cracks and pores of BE through $H_2O_2$ activation, thus reflecting the physical interaction between MNPs and BE phases. Overall, the existence of these spherical-like $Fe_3O_4$ nanoparticles could be projected to enhance the uptake of MB molecules due to the increase of the BE/SD–MNPs surface area. The combination between BE, SD, and MNPs is clarified by TEM images (Figure 2h–j). The BE sample shows the presence of smooth sheets of montmorillonite arranged above each other (Figure 2h). Decoration of the BE/SD interface by MNPs resulted in the loading of BE sheets and fibrous particles of SD by spherical nanoparticles (Figure 2i–k). Therefore, FESEM and TEM results support the modification of BE/SD by $Fe_3O_4$ magnetic nanoparticles.

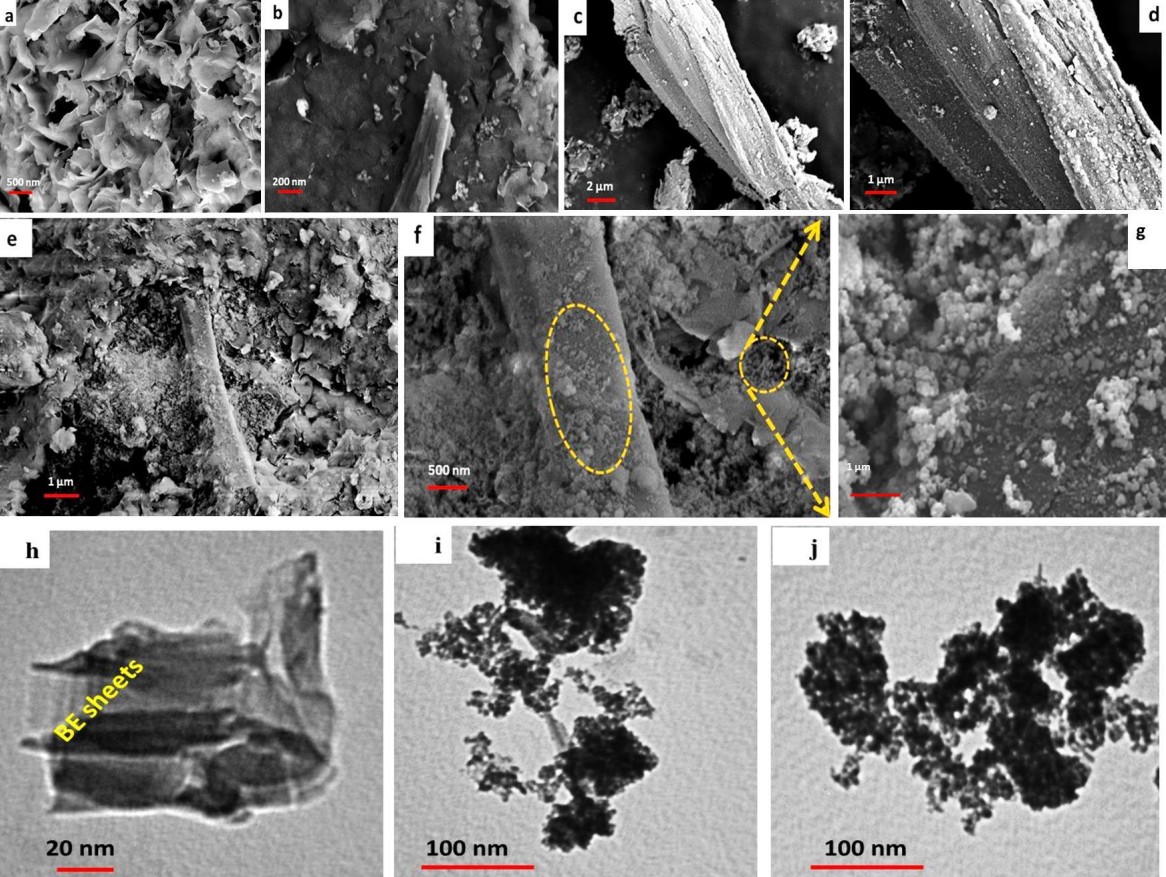

**Figure 2.** Characterization of the studied samples: FESEM images of BE (**a**), BE/SD (**b**–**d**), BE/SD–MNPs (**e**–**g**), TEM images of BE (**h**), and BE/SD–MNPs (**i**,**j**).

An FTIR spectrum of the developed BE/SD–MNPs composite is shown in Figure 3. The detected band at 3626.78 cm$^{-1}$ could be assigned to the hydroxyl groups (–OH) of the octahedral cations, particularly Al$^{3+}$ [30]. The observed band at 3440.10 cm$^{-1}$ is related to the –OH groups of the mechanically held water molecules [31]. The peak detected at 2928.54 cm$^{-1}$ is attributed to the –CH$_2$ stretching of the aliphatic compound [19]. The characteristic band at 2365 cm$^{-1}$ may correspond to an N–H stretching vibration [32]. The detected peak at 1640.53 cm$^{-1}$ could be associated with the bending vibration mode of the attached water molecules [30]. The band located at 1434.06 cm$^{-1}$ could be attributed to Si–O or Al–O groups of the BE sample [18]. The Si–O bending vibration was detected at 1036.59 cm$^{-1}$, while the stretching vibration was observed at 713.72 cm$^{-1}$ [33]. The visible bands at 601.67 and 466.30 cm$^{-1}$ can be associated with the Fe–O vibrational modes, while the band observed at 564.72 cm$^{-1}$ was attributed to Fe–O stretch vibrations of Fe$_3$O$_4$ nanoparticles [5]. Thus, detecting these bands supported the modification of BE/SD by magnetic Fe$_3$O$_4$ nanoparticles producing multifunctional BE/SD–MNPs composite.

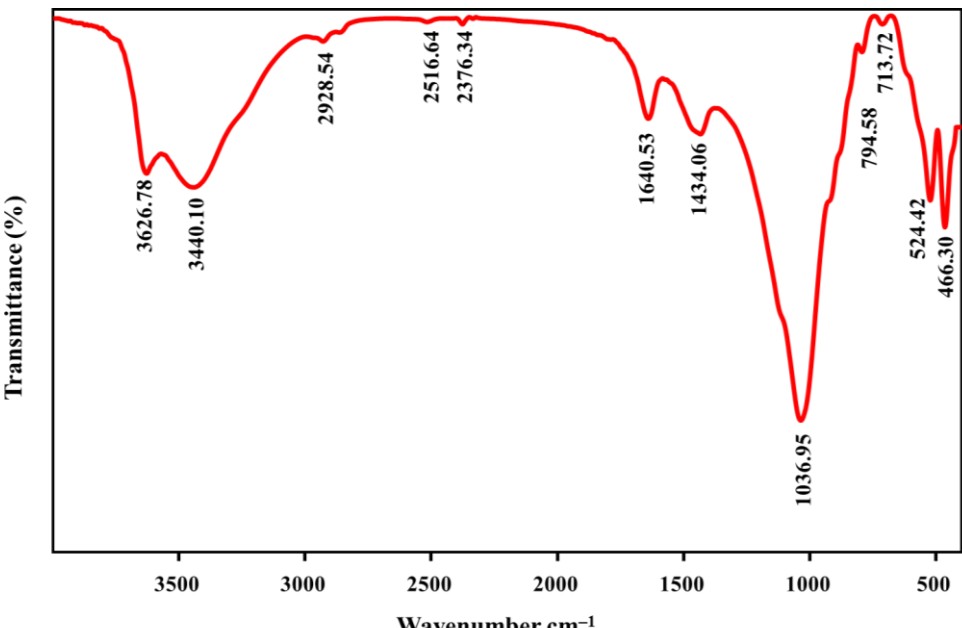

**Figure 3.** FTIR spectrum of the developed BE/SD–MNPs composite.

The TGA curve displays two main weight losses for the investigated BE/SD–MNPs adsorbent detected at different temperatures (Figure 4a). The TGA shows an initial small weight loss (3.28%) at temperatures below 200 °C, which could be associated with the removal of mechanically attached water and the organic content of this composite [34]. In the second, the weight loss occurs at 300–750 °C (5.17%), which may be due to the loss of (-OH) groups in the BE/SD–MNPs structure and to oxygen during Fe$_3$O$_4$ oxidation [35]. Additionally, the DSC curve shows the presence of two endothermic peaks (Figure 4b). The first peak observed at 100–150 °C could be attributed to the evaporation of H$_2$O molecules, while the second one at nearly 600 °C could be due to the liberation of chemically held water in the BE/SD–MNPs structure. Accordingly, the results of the TGA and DSC reflected the thermal stability of BE/SD–MNPs composite at high temperatures.

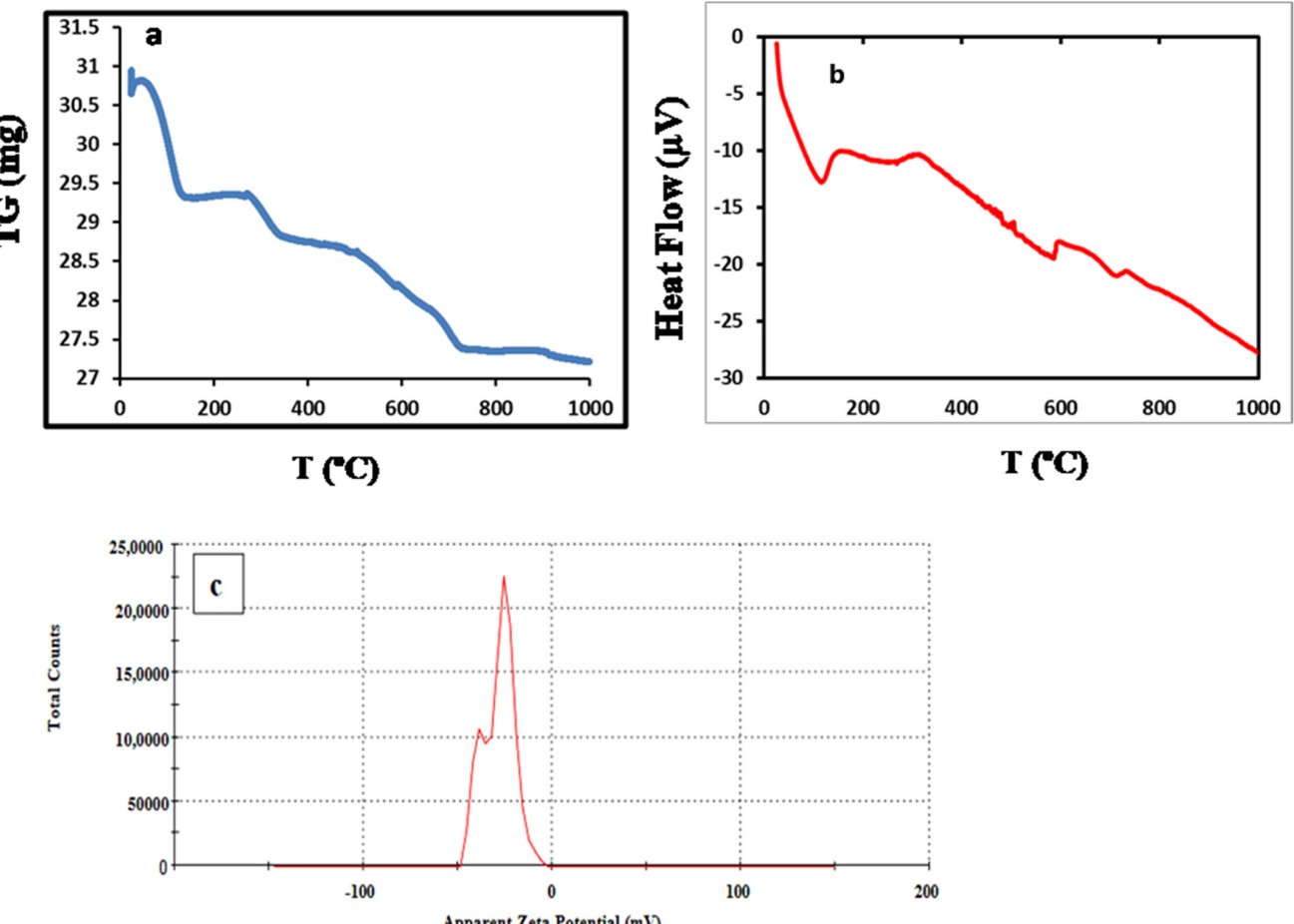

**Figure 4.** TGA (**a**), DSC (**b**), and zeta potential (**c**) results of BE/SD–MNPs.

Zeta potential (ZP) of nanoparticles with values >+25 mV or <−25 mV usually have a high degree of stability [36]. The BE/SD–MNPs composite presented a negative zeta potential of −28.3 mV (Figure 4c). The ZP value becomes more negative with the increment of solution pH due to the increase of hydroxyl ions and deprotonation of the BE/SD–MNPs functional groups. The highly negative charge of the BE/SD–MNPs surface reflected its high stability against aggregation due to the high repulsion forces between the adjacent particles [36]. Consequently, the solution pH 8.0 is selected to carry out all the MB adsorption experiments.

### 3.2. Contact Time Effect on MB Adsorption on BE/SD–MNPs

Three stages were detected during the removal of MB molecules by the BE/SD–MNPs adsorbent (Figure 5a). Stage 1 (i.e., $5 < t < 45$ min) was very fast due to the availability of a significant number of the BE/SD–MNPs active sites for MB adsorption. Stage 2 ($45 < t < 90$ min) displayed a gentle slope, which could be related to the intra-particle diffusion at which the adsorbed amount of MB increased from $134.5 \pm 0.48$ to $144.2 \pm 0.88$ mg/g, signifying a small MB dye uptake in this stage of contact time. In the last stage, the removed amounts of MB were nearly constant, reflecting the saturation of the BE/SD–MNPs active sites by the adsorbed MB molecules (i.e., equilibrium state).

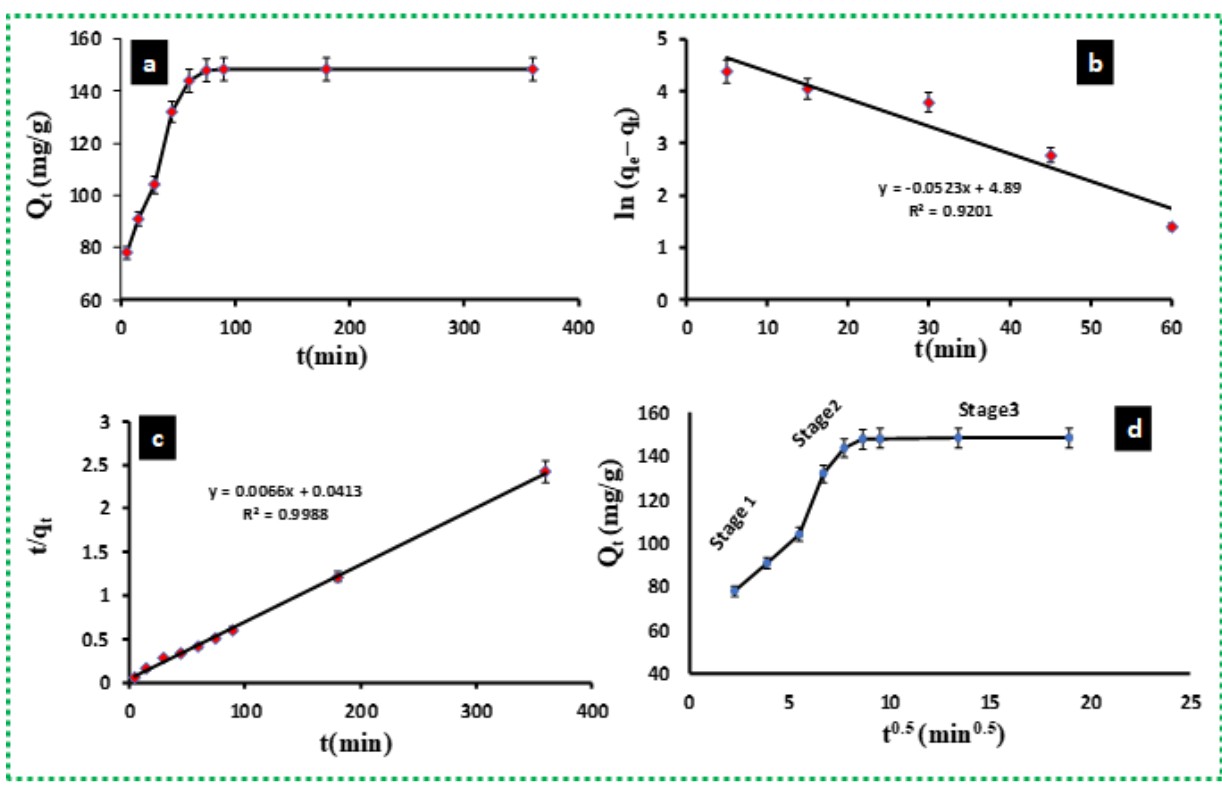

**Figure 5.** Effect of contact time (**a**), the pseudo-first-order (**b**), the pseudo-second-order (**c**), and intra-particle diffusion (**d**) equations associated with the adsorption of MB on BE/SD–MNPs. Operating conditions: 25 mg of BE/SD– MNPs mass, 25 mL of M.B. solution (150 mg/L), 25 °C of adsorption temperature, and solution pH 8.0.

*3.3. MB Adsorption Kinetics*

The plots of $\ln(q_e - q_t)$ versus $t$ (Figure 5b) as well as $t/q_t$ against $t$ (Figure 5c) were used in finding the parameters associated with the pseudo-first-order (PFO) and the pseudo-second-order (PSO) equations, respectively. The parameters' results from the PFO and PSO kinetics models are listed in Table 1.

**Table 1.** Parameters of kinetics models for MB uptake by the BE/SD–MNPs composite.

| Kinetics Models | Parameters | | $R^2$ |
|---|---|---|---|
| Pseudo-first-order | $q_e$ (mg/g) | $k_1$ (1/min) | |
| | 151.51 | 0.0523 | 0.9201 |
| Pseudo-second-order | $q_e$ (mg/g) | $k_2$ (g/mg min) | |
| | 19.12 | 0.00178 | 0.9988 |
| Intra-particle diffusion | $k_p$ [(mg/g) (1/min) 0.5)] | $C$ (mg/g) | |
| | 4.2001 | 91.180 | 0.5789 |

Based on the $R^2$ values (Table 1), the PSO equation ($R^2$ = 0.9988) described the MB adsorption process well compared to the PFO ($R^2$ = 0.9201). Additionally, the calculated and experimental $q_e$ values were nearly equivalent, reflecting the suitability of the PSO in fitting the MB adsorption data. Thus, the whole rate of MB removal by the BE/SD–MNPs was mainly related to the chemical interface between the dye molecules and the active adsorbent sites.

### 3.4. Diffusion Mechanism

The relation between $q_t$ and $t^{1/2}$ (Figure 5d) displayed a multi-linear plot (i.e., different linear stages) of the MB experimental data, which reflected different diffusion types. The first sharp stage explains the external mass transfer of MB from the contaminated solution to the outside surface of the developed BE/SD–MNPs composite. The second and last stages reflect the pore-diffusion and equilibrium phases, respectively. Consequently, the adsorption of MB molecules on the BE/SD–MNPs composite was directed by more than one mechanism (i.e., chemical reaction and pore-diffusion processes were involved in the uptake of MB dye by the tested adsorbent).

### 3.5. Classical and Advanced Modeling of MB Adsorption on BE/SD–MNPs

The experimental data achieved through the interface between MB molecules and BE/SD–MNPs active sites were fitted to traditional and advanced statistical physics models. The Wolfram Mathematica 10 program and statistical metrics were utilized to analyze the attained adsorption results. The physicochemical parameters associated with the applied models were analyzed and discussed to evaluate the BE/SD–MNPs performance, and the adsorption mechanism is provided in the next sections.

#### 3.5.1. Traditional Isotherm Models

The changeable parameters attained from modeling MB adsorption isotherms utilizing the non–linear form of the Langmuir and Freundlich models are shown in Figure 5 and Table 2. The values of $R^2$ revealed that MB adsorption onto the BE/SD–MNPs composite was described by both the Langmuir and Freundlich models at 25, 40, and 55 °C ($R^2 > 0.99$). The maximum MB adsorption capacities ($q_{max}$) were 347.99, 385.27, and 388.12 mg/g at 25, 40, and 55 °C, respectively. (Table 2). Consequently, the uptake of MB by the BE/SD–MNPs composite improved with the increment of adsorption temperature (i.e., MB adsorption was controlled by its endothermic nature). Furthermore, the $K_F$ value of the Freundlich model also increased from 7.24 to 14.61 mg/g by enhancing the solution temperature from 25 to 55 °C. The behavior of $K_F$ with temperature supported by the endothermic nature of MB adsorption on the BE/SD–MNPs. Additionally, the $1/n$ values (Table 1) were within the range of 0.58–0.62 (i.e., <1.0), thus signifying a positive MB removal at low concentrations [28]. Overall, to find the best-fitted model, the values of $\chi^2$ presented in Table 1 were used. The $\chi^2$ values of the Freundlich model were lower than that of the Langmuir model at the same temperatures (i.e., Freundlich is the best model). Therefore, heterogeneous functional groups of the BE/SD–MNPs composite were responsible for MB adsorption.

**Table 2.** Parameters of isotherms models for MB removal by the BE/SD–MNPs composite.

| Isotherm Model | 25 °C | 40 °C | 55 °C |
|---|---|---|---|
| Langmuir | | | |
| $q_{max}$ (mg/g) | 347.99 | 385.27 | 388.12 |
| $k_L$ (L /mg) | 0.00634 | 0.00776 | 0.01266 |
| $R^2$ | 0.9984 | 0.9985 | 0.9966 |
| $\chi^2$ | 2.258 | 2.121 | 4.639 |
| Freundlich | | | |
| $k_F \left( (mg/g)(mg /L)^{-1/n} \right)$ | 7.24 | 9.57 | 14.61 |
| $1/n$ | 0.62 | 0.61 | 0.58 |
| $R^2$ | 0.9989 | 0.9975 | 0.9946 |
| $\chi^2$ | 1.547 | 1.203 | 3.992 |

Different natural and developed materials (e.g., clays, zeolite, diatomite, graphene, rice husk, and $Fe_3O_4$/montmorillonite) were utilized as adsorbents for MB uptake from solutions (Table 3). Nevertheless, results of the current study indicated that the BE/SD–MNPs composite presented high $q_{max}$ values compared to the other adsorbents listed in Table 3. Therefore, the prepared BE/SD–MNPs composite is recommended to be a favorable and low-cost adsorbent for the remediation of MB-bearing solutions.

**Table 3.** MB adsorption capacities of the BE/SD–MNPs composite and different adsorbents.

| Adsorbent | Sorption Capacity (mg/g) | Reference |
|---|---|---|
| Graphene | 153 | [37] |
| Kaolin | 45 | [38] |
| Montmorillonite | 64 | [39] |
| Chitosan/magnetic silica | 201 | [40] |
| Zeolite 4A | 22 | [38] |
| Polydopamine microspheres | 90.7 | [12] |
| Activated rice husk | 65 | [41] |
| Ball clay | 25 | [13] |
| $Fe_3O_4$/montmorillonite | 69 | [42] |
| Rt/BC | 214.52 | [22] |
| Fibrous clay minerals | 39–85 | [43] |
| Purified diatomite | 105 | [44] |
| $Fe_3O_4$/serpentine composite | 201 | [21] |
| Activated carbon | 289.25 | [45] |
| BE/SD–MNPs composite | 347.99 | This study |

3.5.2. Statistical Physics Models

Moreover, applying the statistical physics theory is a confident technique for understanding the interaction mechanism between the captured MB molecules and the active sites of the BE/SD–MNPs adsorbent [17,45]. Overall, MB molecules and BE/SD–MNPs interaction was primarily directed by steric (i.e., $n$, $D_M$, $N_t$, and $Qsat$) and energetic ($\Delta E$) parameters of the selected statistical model. Interpretation of these physicochemical factors can evidently clarify the MB adsorption mechanism at the molecular scale. The greater $R^2$ and the lowest RMSE values reflected that the multilayer model (M3) is acceptable for signifying the MB adsorption results at all temperatures compared to M1 and M2. (Table 4 and Figure 6). Therefore, the physicochemical parameters of M3 were calculated and deeply interpreted at all adsorption temperatures as illustrated below.

**Table 4.** $R^2$ and RMSE values for the tested advanced isotherms models for MB uptake by the BE/SD–MNPs composite.

| Statistical Model | T (°C) | Parameters | |
|---|---|---|---|
| | 25 | $R^2$ | 0.9978 |
| | | RMSE | 2.613 |
| **Monolayer (M1)** | 40 | $R^2$ | 0.9976 |
| | | RMSE | 2.596 |
| | 55 | $R^2$ | 0.9959 |
| | | RMSE | 3.747 |

**Table 4.** *Cont.*

| Statistical Model | T (°C) | Parameters | |
|---|---|---|---|
| **Double layer (M2)** | 25 | $R^2$ | 0.9970 |
| | | RMSE | 3.487 |
| | 40 | $R^2$ | 0.9931 |
| | | RMSE | 9.698 |
| | 55 | $R^2$ | 0.9969 |
| | | RMSE | 3.014 |
| **Multilayer (M3)** | 25 | $R^2$ | 0.9991 |
| | | RMSE | 0.9834 |
| | 40 | $R^2$ | 0.999 |
| | | RMSE | 1.426 |
| | **55** | $R^2$ | 0.9964 |
| | | RMSE | 2.701 |

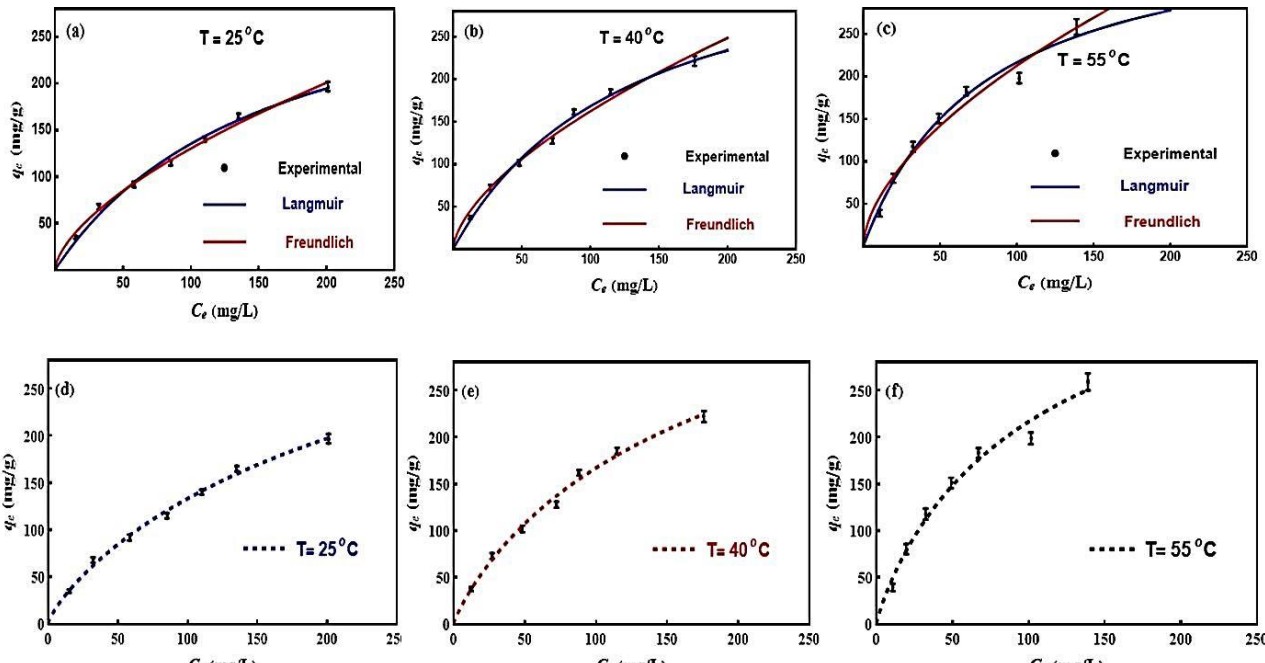

**Figure 6.** Langmuir, Freundlich, and multilayer adsorption models utilized for fitting the MB uptake by the BE/SD–MNPs composite. Classical models (**a–c**) and advanced monolayer model (**d–f**) at different temperatures.

*3.6. Steric Parameters Interpretation*

3.6.1. The *n* Parameter

The values of parameter *n* can offer deep insights into the interaction mechanism of MB dye molecules on the BE/SD–MNPs surface. This parameter can define the geometry and mechanism of MB molecules removed by the BE/SD–MNPs active sites. Furthermore, this steric parameter can provide the effect of temperature on changing the adsorption mechanism. Overall, three probable cases can describe the MB adsorption orientation according to this parameter [2,5,17,22,45].

Case 1: *n* < 0.5: This condition indicates that the BE/SD–MNPs functional group can remove a portion of dye molecules (i.e., MB molecule be captured by two or more active sites of BE/SD–MNPs), presenting a parallel adsorption geometry and multi-docking uptake mechanism.

Case 2: 0.5 < *n* < 1: According to this situation, MB molecules can be removed through parallel and non-parallel orientations with varying percentages (i.e., a mixed geometry).

Case3: $n \geq 1$: Based on this state, the functional group of BE/SD–MNPs can accept one or more MB molecules, thus suggesting a non-parallel orientation (i.e., a multi-molecular uptake mechanism).

Calculations attained from the M2 displayed that the removed number of MB molecules per adsorption site ($n$) increased from 0.75 to 0.85 in the temperature range of 25–55 °C (Table 5). Therefore, the $n$ parameter values followed the second case (i.e., $0.5 < n < 1$) at all adsorption temperatures. This behavior reflected that the interactions between MB molecules and BE/SD–MNPs active sites were considered by parallel and non-parallel orientations (i.e., horizontal and vertical positions for MB molecules were identified). In addition, multi–docking and multi-molecular mechanisms were also involved during the adsorption process. The detection of mixed geometry and mechanism could be related to the presence of different functional groups of BE/SD–MNPs with diverse adsorption energies involved in the uptake of this cationic dye. The variation of $n$ as a function of temperature is displayed in Figure 7. It can be concluded that the solution temperature has no clear impact in altering the geometry and mechanism associated with the MB and BE/SD–MNPs interaction.

**Table 5.** Steric and energetic parameters of the advanced multilayer model for MB uptake by the BE/SD–MNPs composite.

| $T$ (°C) | $n$ | $D_M$ (mg/g) | $1 + N_2$ | $Q_{sat}$ (mg/g) | $\Delta E_1$ (kJ/mol) | $\Delta E_2$ (kJ/mol) |
|---|---|---|---|---|---|---|
| **25** | 0.75 | 482.73 | 2.34 | 829.10 | 14.30 | 11.62 |
| **40** | 0.82 | 464.16 | 2.21 | 845.75 | 15.44 | 13.90 |
| **55** | 0.85 | 458.24 | 2.19 | 848.99 | 16.04 | 15.92 |

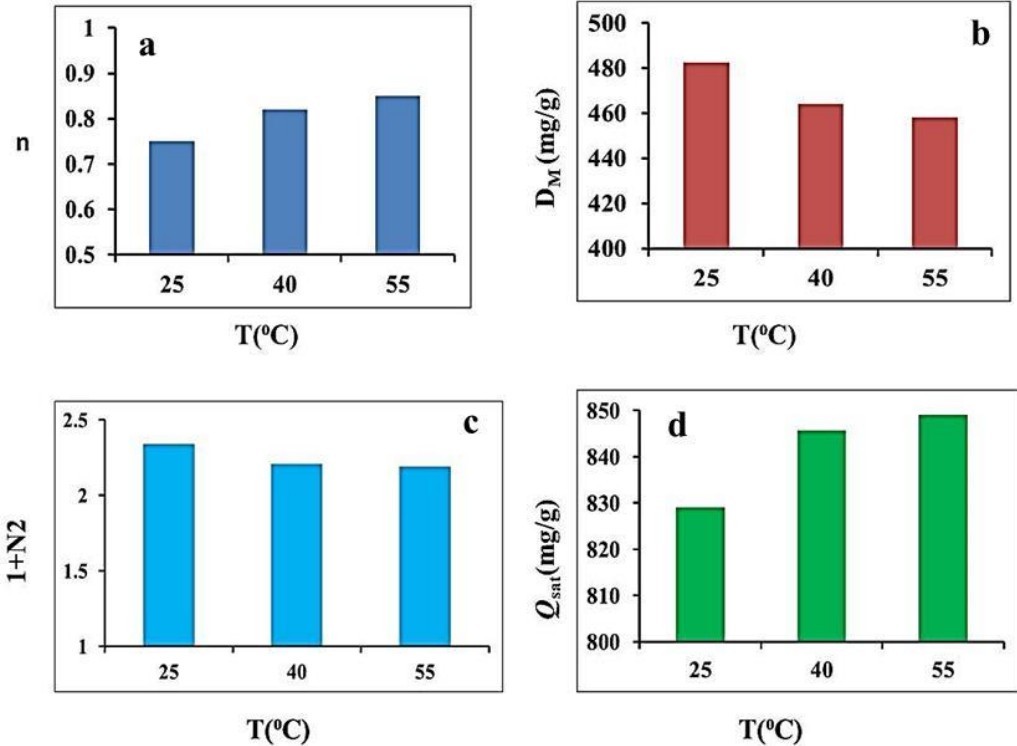

**Figure 7.** The behavior of the number of MB molecules per site (**a**), the density of receptor sites (**b**), the formed number of MB layers (**c**), and adsorption capacity at saturation (**d**) as a function of temperature for MB adsorption on the BE/SD–MNPs composite.

### 3.6.2. The $D_M$ Parameter

The behavior of this parameter against temperature for the MB adsorption onto the BE/SD–MNPs composite systems is presented in Table 5 and Figure 7. This steric parameter displayed a decrease of the $D_M$ value from 482.73 to 458.24 within the temperature 25–55 °C. Overall, the adsorption temperature has an opposite effect on the $D_M$ as compared to the $n$ parameter (i.e., with increasing temperature the $n$ value increased and the $D_M$ value decreased). This reversal of behavior between the two steric parameters ($n$ and $D_M$) was reported in previous studies [2,5,17,45]. Usually, the MB–MB interface increased the n parameter and, therefore, decreased the number of the occupied BE/SD–MNPs active sites (i.e., a reduction of the density of the adsorbent functional groups). Accordingly, the increment of temperature resulted in the hidden of some functional groups of BE/SD–MNPs that were involved in MB uptake at lower temperatures.

### 3.6.3. Total Number of Formed Layers ($Nt = 1 + N_2$)

The trend of this parameter with changing the adsorption temperature from 25 to 55 °C is shown in Figure 7, and the results are listed in Table 5. The total number of the adsorbed MB layers was 2.34, 2.21, and 2.19 at 25, 40, and 55 °C, respectively. The main reason associated with the decrease of this value could be due to the thermal agitation effect, which decreased the attraction forces between the adsorbed MB molecules (i.e., a disordered movement of the removed MB dye) in the multilayer state [17]. Furthermore, the slight difference in the $N_t$ at all temperatures reflected the insignificant role of this parameter in governing the interaction between MB molecules and BE/SD–MNPs adsorption sites.

### 3.6.4. Adsorbed MB Quantity at Saturation ($Q_{sat}$ Parameter)

Determination of the adsorption capacity at saturation [$Q_{sat} = n. DM (1 + N_2)$] is a vital issue for evaluating the uptake efficiency of the BE/SD–MNPs adsorbent for capturing MB molecules. The $Q_{sat}$ values were 829.10, 845.75, and 848.99 mg/g at temperatures of 25, 40, and 55 °C, respectively. (Table 5 and Figure 7). The increase of the $Q_{sat}$ values with temperature supported the endothermic interaction between the MB molecules and BE/SD–MNPs. The values of $Q_{sat}$ displayed the same style presented by the $n$ parameters (i.e., the $N_2$ and $D_M$ parameters increased with temperature). (Table 5). This behavior suggested that the steric $n$ parameter mainly controlled the BE/SD–MNPs adsorption performance. Similar results were reported in our previous study [28]. Furthermore, if it was expected that $N_2$ = zero, the results of $n. DM$ presented very similar values to the monolayer adsorption model, reinforcing the correspondence between the experimental and theoretical adsorption data.

### 3.7. Interpretation of MB Adsorption on BE/SD–MNPs via Adsorption Energy ($\Delta E$)

The selected model (M3) suggests the existence of two adsorption energies ($\Delta E_1$ and $\Delta E_2$) that can be involved in MB adsorption on the BE/SD–MNPs active sites. The values of ($\Delta E_1$ and $\Delta E_2$) were determined by Equations (19) and (20) as follows [2,17,45]:

$$C_1 = C_s e^{-\frac{\Delta E_1}{RT}} \tag{19}$$

$$C_2 = C_s e^{-\frac{\Delta E_2}{RT}} \tag{20}$$

where $c_1$ and $c_2$ signify the two concentrations at half-saturation and $c_s$ represents the MB solubility in water.

Figure 8 and Table 5 display the values of MB adsorption energies concerning the solution temperature. At all operating temperatures, the $\Delta E_1$ and $\Delta E_2$ presented positive values, which confirmed the endothermic nature of the interaction between the MB and BE/SD–MNPs. Overall, the adsorption energies were <40 kJ/mol, which reflected the involvement of the physical forces (e.g., hydrogen bindings, van der Waals, and electrostatic interactions) in MB dye and the BE/SD–MNPs interface. As predictable, the values of

$\Delta E_1$ were greater than those of $\Delta E_2$ at the three temperatures because $\Delta E_1$ was associated with the direct interface between MB and the adsorbent surface, while $\Delta E_2$ reflected the MB–MB interactions. Energetically, the $\Delta E$ style was comparable with that of the $Q_{sat}$ with increasing temperature. Consequently, the MB adsorption mechanism on BE/SD–MNPs was mainly directed by the steric ($n$) and energetic ($\Delta E$) parameters.

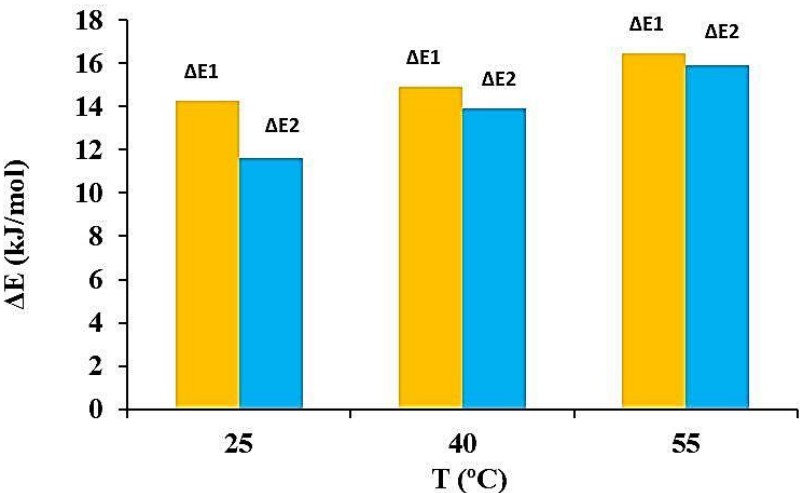

**Figure 8.** The behavior of energetic parameters as a function of temperature for MB adsorption. by BE/SD–MNPs composite.

Figure 9 displays the geometry and adsorption mechanism of MB molecules on the BE/SD–MNPs adsorbent based on the multilayer statistical physics model.

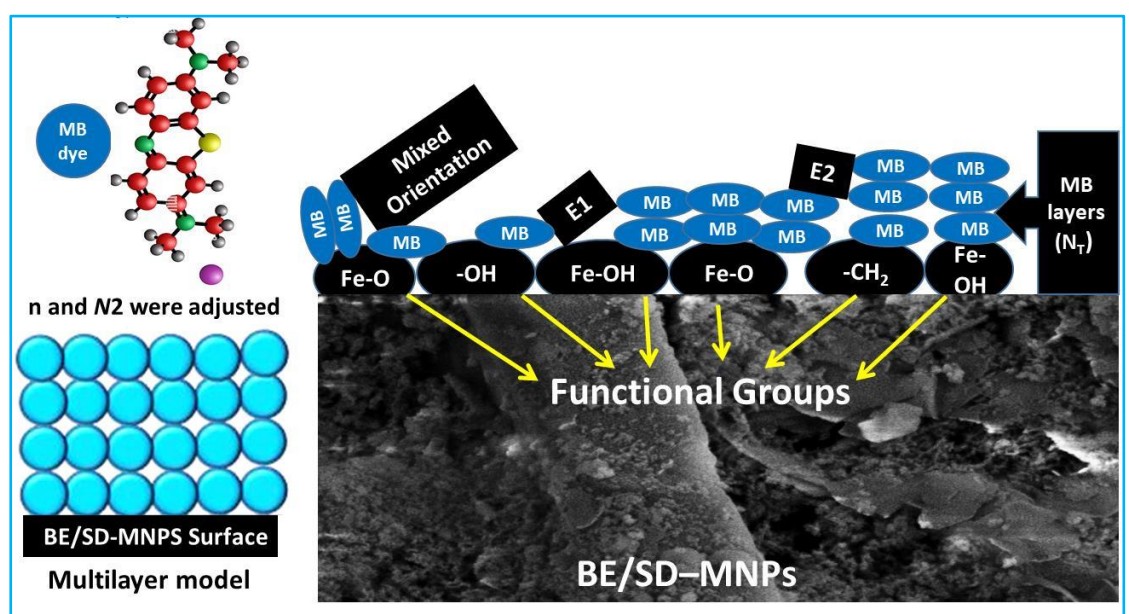

**Figure 9.** The geometry and adsorption mechanism of MB on the BE/SD–MNPs adsorbent via the multilayer model.

*3.8. Regeneration of Rt/BC Adsorbent*

After all the regeneration runs, the as-prepared BE/SD–MNPs composite presented removal percentages [%] above 85% for MB (Figure 10). The slight decrease in sorption capacities can be related to two possible factors: (i) some amounts of MB dye could not be desorbed (i.e., irreversible uptake); and (ii) slow removal of porosity and surface chemistry

upon processing [21]. Therefore, the BE/SD–MNPs adsorbent can be recycled many times by removing MB dye without a significant loss in its uptake efficiency, representing the high stability and recyclability of the BE/SD–MNPs composite in industrial systems.

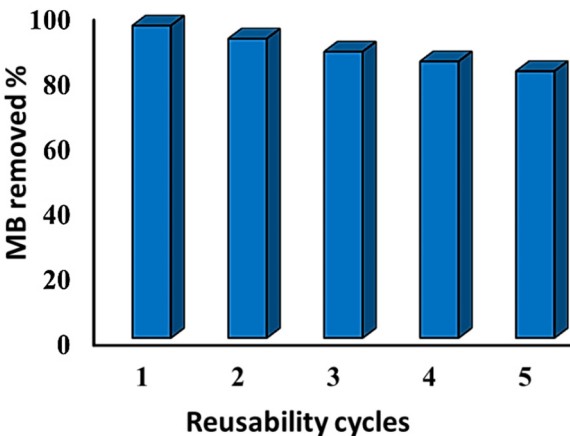

**Figure 10.** Percentage of MB uptake during the adsorption/desorption cycles.

*3.9. Economic Profit*

Generally, using low-cost natural materials and solid wastes in fabricating effective adsorbents for capturing water pollutants is highly recommended. Concerning the components of the tested BE/SD–MNPs composite, clays are inexpensive and available natural materials, wood sawdust is locally accessible with huge amounts, and Fe-bearing materials such as weathered basalt or fired brick wastes are very reachable. Overall, this study provides a distinguishing method that takes the available raw materials, industrial solid wastes, and Fe-rich natural minerals or wastes to the next level in producing multifunctional composites utilized in the remediation of dyes-bearing water. In addition, the costs of the water remediation process can be decreased substantially if the employed adsorbents can be recycled and reused for numerous cycles as presented by the BE/SD–MNPs composite of the current study.

**4. Conclusions**

A highly efficient BE/SD-MNPs composite was synthesized successfully and used for the removal of the methylene blue dye. The dye adsorption experimental data were fitted to classical and statistical physics equilibrium models. The Freundlich equation and a multilayer model with saturation fitted the adsorption results at all tested temperatures. Steric and energetic parameters associated with the advanced multilayer model were calculated and interpreted. According to the *n* parameter, the MB molecules were separated in solutions offering a mixture of vertical and horizontal adsorption orientations onto BE/SD–MNPs active sites, increasing the adsorption capacity at saturation ($Q_{sat}$) from 829 to 849 mg/g within the temperature range of 25–55 °C, reflecting endothermic reactions. Adsorption energies related to MB uptake were less than 40 kJ/mol, signifying physical and endothermic interactions. Regeneration results clarified that the BE/SD-MNPs reserved >85% of MB uptake after five adsorption-desorption cycles. These theoretical results offered new insights and a suitable understanding of the adsorption mechanism of MB on the practical and low-cost multifunctional BE/SD-MNPs composite.

**Author Contributions:** Methodology, M.K.S., software, R.F.H.; writing—original draft preparation, M.A.B., R.K. and B.A.A.-M., writing—review and editing, M.K.S., R.K. and M.A.B. All authors have read and agreed to the published version of the manuscript.

**Funding:** This article was financially supported by the Deanship of Scientific Research (DSR) at King Abdulaziz University, Jeddah, under grant no G: 149-155-1443.

**Acknowledgments:** The authors acknowledge with thanks the DSR for technical and financial support.

**Conflicts of Interest:** The authors declare no conflict of interest.

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
