# Peer review of "Fe3O4 Nanoparticles Loaded Bentonite/Sawdust Interface for the Removal of Methylene Blue: Insights into Adsorption Performance and Mechanism via Experiments and Theoretical Calculations"

_water, doi:10.3390/w14213491_

Round 1

Reviewer 1 Report

Dear Authors,

The aim of the "Fe3O4 nanoparticles loaded bentonite/sawdust interface for the removal of methylene blue: Insights into adsorption performance and mechanism via experiments and theoretical calculations" study is well explained, describing the fabrication and characterization of the magnetic clays for use in wastewater remediation.

The manuscript should be written using the journal's template, and I highly recommend an extensive English revision. Now, the paper is hard to read because it does not have the same style and format, is more space after rows and in the tables. I highly recommend that the authors correct the names of the substances in the whole manuscript (for example, in row 108: NH4OH must be written NH4OH, 4- with subscript).

The authors must improve the quality and explanations of the figures and tables. In figure 1, under the first beaker, I recommend writing its content and adding it to the arrow "+ H2O2". Figure 3 is too bigger; In Figure 4, the graphs do not have the same format, and graph c has a low quality, in Figure 6 is not write the explanations for graphs from a to f. Also, in Fig. 7, all four graphs are not numbered and explained.

The experiment from the 3.8 section may be performed in triplicate, and in Fig. 10, the data must be interpreted from a statistical point of view.

Furthermore, all references need to be formatted using the journal's template.

Author Response

Responses to Reviewers

The authors of this article sincerely thank the reviewer for his useful comments and the time spent suggesting revising this manuscript. All the suggestions of the reviewer have been carefully considered, and we have tried our best to improve the manuscript and the corresponding corrections have been added to the revised manuscript. This text was prepared to present our point-by-point responses to the comments of the reviewer. The full corrections are achieved in red color as the following:

Reviewer 1

Comments and Suggestions for Authors

Dear Authors,

The aim of the "Fe3O4 nanoparticles loaded bentonite/sawdust interface for the removal of methylene blue: Insights into adsorption performance and mechanism via experiments and theoretical calculations" study is well explained, describing the fabrication and characterization of the magnetic clays for use in wastewater remediation.

The manuscript should be written using the journal's template, and I highly recommend an extensive English revision. Now, the paper is hard to read because it does not have the same style and format, is more space after rows and in the tables. I highly recommend that the authors correct the names of the substances in the whole manuscript (for example, in row 108: NH4OH must be written NH4OH, 4- with subscript).

Response: Thanks very much for this comment: The manuscript was written using the journal's template, and the article was checked carefully using an extensive English revision. All figures and tables were improved. Also, the names of the substances in the whole manuscript were corrected. Please see the updated article.

The authors must improve the quality and explanations of the figures and tables. In figure 1, under the first beaker, I recommend writing its content and adding it to the arrow "+ H2O2". Figure 3 is too bigger; In Figure 4, the graphs do not have the same format, and graph c has a low quality, in Figure 6 is not write the explanations for graphs from a to f. Also, in Fig. 7, all four graphs are not numbered and explained.

Response: Thanks very much for this comment: The quality and explanations of figures and tables was improved, Figures 3 and 4 were modified. In figure 6, the explanations for graphs from a to f were added. In figure 7,  the four graphs were numbered and explained, please see figures and tables in the corrected article.

The experiment from the 3.8 section may be performed in triplicate, and in Fig. 10, the data must be interpreted from a statistical point of view.

Response: Thanks very much for this comment , all MB adsorption experiments were repeated three times, and the mean values of the results have been utilized for data evaluation, with the error being always below ± 5 %. In this study, we focused on understanding the MB adsorption mechanism at the molecular scale through the application of the statistical physics theory. So, we request the Editor and reviewer to kindly accept our article without this information.

Furthermore, all references need to be formatted using the journal's template.

Response: Thanks very much for this comment, all references were formatted. Please see the corrected article.

Reviewer 2 Report

Current study “Fe3O4 nanoparticles loaded bentonite/sawdust interface for the removal of methylene blue: Insights into adsorption performance and mechanism via experiments and theoretical calculations” basically based on kinetic, equilibrium and thermodynamic studies. Instead of using these common terminologies author used theoretical, modeling and energies etc. Current results are structured nicely but lacking in many aspects of adsorption studies. Before going towards kinetic, equilibrium and thermodynamic studies, comprehensive optimization study to optimize all the experimental or operational parameters like pH, concentrations of dye, different doses of adsorbents, temperature etc. is required.  Author used single values of all these parameters. Discussion of results are insufficient. I suggest major revisions are required to upgrade the quality of manuscript.  

Page 2: Lines 75-88 started from “The physicochemical features of the prepared BE/SD–MNPs composite….” (delete this portion).

The size or diameter of nanoparticles were not included in FESEM results. Why?

Surface area of the nanoparticles are necessary for adsorption study, please include BET study to determine surface area of the nanoparticles.

Why you select single pH 8.0 value for all experiments, also single value for MB concentration 150 mg/L?

Why you select single dose of adsorbent 25 mg?

Page 9: line 284 correct DCS to DSC.

Author Response

Responses to Reviewers

The authors of this article sincerely thank the reviewer for his useful comments and for the time spent on suggesting for revising this manuscript. All the suggestions of the reviewer have been carefully considered, and we have tried our best to improve the manuscript and the corresponding corrections have been added to the revised manuscript. This text was prepared to present our point-by-point responses to the comments of the reviewer. The full corrections are achieved in red color as the following:

Reviewer 2

Comments and Suggestions for Authors

Current study, “Fe3O4 nanoparticles loaded bentonite/sawdust interface for the removal of methylene blue: Insights into adsorption performance and mechanism via experiments and theoretical calculations” basically based on kinetic, equilibrium and thermodynamic studies. Instead of using these common terminologies author used theoretical, modeling and energies etc. Current results are structured nicely but lacking in many aspects of adsorption studies. Before going towards kinetic, equilibrium and thermodynamic studies, a comprehensive optimization study to optimize all the experimental or operational parameters like pH, concentrations of dye, different doses of adsorbents, temperature etc. is required. The author used single values of all these parameters. Discussion of results are insufficient. I suggest major revisions are required to upgrade the quality of the manuscript. 

  • Page 2: Lines 75-88 started from “The physicochemical features of the prepared BE/SD–MNPs composite….” (delete this portion).
  • Response: Thanks very much for this comment: The following portion: The physicochemical features of the prepared BE/SD–MNPs composite were investigated using several techniques, including FTIR, FESEM, TEM, TGA, DSC, and Zeta potential was deleted, please see the corrected article.
  • The size or diameter of nanoparticles were not included in FESEM results. Why?
  • Response: Thanks very much for this comment: The diameter of nanoparticles was included as follows: Moreover, BE/SD-supported by Fe3O4 nanoparticles (Fig. 2e–g) displays the decoration of BE/SD by the agglomerated spherical magnetic nanoparticles with dissimilar diameters less than 100 nm forming the BE/SD–MNPs composite.
  • Surface area of the nanoparticles are necessary for adsorption study, please include BET study to determine surface area of the nanoparticles.
  • Response: Actually, some techniques are not available in the institutes and universities of the research group, thus including the BET surface area measurement. To carry out this analysis, more than 60 days can be required. So, we request the Editor and reviewer to kindly accept our article without this information.

  • Why you select single pH 8.0 value for all experiments, also single value for MB concentration 150 mg/L?
  • Response: Thanks very much for this comment. BE/SD–MNPs composite presented a negative zeta potential of −28.3 mV (Fig. 4c). The ZP value becomes more negative with the increment of solution pH due to the increase of hydroxyl ions and deprotonation of the BE/SD–MNPs functional groups. The highly negative charge of the BE/SD–MNPs surface reflected its high stability against aggregation due to the high repulsion forces between the adjacent particles [36]. Consequently, the solution pH 8.0 is selected to carry out all the MB adsorption experiments. Also at pH 8.0, the studied composite presented high performance as compared to that of pHs 3.0, 5.0, and 7.0. Furthermore, we used a single value for MB concentration (150 mg/L) to evaluate only the effect of contact time for optimization. Also, the aim of this study was understand the MB adsorption mechanism and the performance of BE/SD–MNPs using the advanced statistical physics models. Therefore, MB concentration was varied from 50 to 400 mg/L at three solution temperatures. Consequently, we request the Editor and reviewer to kindly accept our article without this information.

  • Why you select single dose of adsorbent 25 mg?

Response: Thanks very much for this comment. In this study we did not study the effect of adsorbent mass in the adsorption process and thus, the adsorbent (BE/SD–MNPs) mass was kept at 25 mg  was changing the shaking time (in kinetics study)  as well as the concentration (in isotherms study). Meanwhile, the aim objective of this study was understand the MB adsorption mechanism and the performance of  BE/SD–MNPs at different adsorption temperatures via the advanced statistical physics models.

  • Page 9: line 284 correct DCS to DSC.
  • Response: Thanks very much for this comment. The DCS was changed to DSC, please see the corrected article.

Reviewer 3 Report

The authors report the Fe3O4 nanoparticles loaded bentonite/sawdust interface for the 2 removal of methylene blue: Insights into adsorption performance and mechanism via experiments and theoretical calculations.

I have some observations to make.

1.    The aim of the study is not clearly defined in the abstract

2.    Page 04, line 147-152, Why did the authors choose to work at ph8? Why also they were satisfied with a single initial concentration of 25mg/l. How did they draw the isotherms? 04 initial concentrations are needed.

3.    Page 09, Fig 3, In my opinion it is better to plot the three infrared diagrams on the same figure, so that we can see the difference between brick alone, bentonite alone and the composite

4.    Page 10-11section « MB adsorption kinetics », the authors must mention at which initial concentration they found these values of kinetic constant, in addition the adsorption kinetics is followed at several initial concentrations of the pollutant

5.    Fig 05, authors must add the operating conditions such as: temperature, s/l ratio, time, pH, etc.

6.    The authors should determine the specific surface area by BET of their materials.

7.    The quality of all figures need to be impoved

8.    Why has the isoelectric pH not been determined, since it allows to set the ph of the work ?

9.    All Figures: standard deviations should be given.

Author Response

Responses to Reviewers

The authors of this article sincerely thank the reviewer for his useful comments and for the time spent on suggesting for revising this manuscript. All the suggestions of reviewer have been carefully considered, and we have tried our best to improve the manuscript and the corresponding corrections have been added to the revised manuscript. This text was prepared to present our point-by-point responses to the comments of the reviewer. The full corrections are achieved in red color as the following:

Reviewer 3

Comments and Suggestions for Authors

The authors report the Fe3O4 nanoparticles loaded bentonite/sawdust interface for the 2 removal of methylene blue: Insights into adsorption performance and mechanism via experiments and theoretical calculations.

I have some observations to make.

  1. The aim of the study is not clearly defined in the abstract.

1-Response: Thanks very much for this comment. The aim of this study was clarified in abstract as follows:  Overall, the aim of this study was understand the MB adsorption mechanism using a magnetic clays/lignocellulosic interface as a promising strategy in wastewater remediation, please see the corrected article.

  1. Page 04, line 147-152, Why did the authors choose to work at ph8? Why also they were satisfied with a single initial concentration of 25mg/l. How did they draw the isotherms? 04 initial concentrations are needed.

2- Response: Thanks very much for this comment. BE/SD–MNPs composite presented a negative zeta potential of −28.3 mV (Fig. 4c). The ZP value becomes more negative with the increment of solution pH due to the increase of hydroxyl ions and deprotonation of the BE/SD–MNPs functional groups. The highly negative charge of the BE/SD–MNPs surface reflected its high stability against aggregation due to the high repulsion forces between the adjacent particles [36]. Consequently, the solution pH 8.0 is selected to carry out all the MB adsorption experiments. Also at pH 8.0, the studied composite presented high performance as compared to that of pHs 3.0, 5.0, and 7.0.

To evaluate the kinetics studies of MB adsorption onto BE/SD–MNPs, the other parameters (i.e., adsorbent mass, MB concentration, solution pH, and temperature) were kept constant.  Thus, the variation of BE/SD–MNPs performance was mainly related to changing the contact time from 5 min to 360 min.

Regarding the isotherms study, the initial MB concentrations were changed between 50 mg/L to 400 g/L as we reported that in the original article in section: 2.5. Equilibrium studies of MB adsorption onto BE/SD–MNPs.

  1. Page 09, Fig 3, In my opinion it is better to plot the three infrared diagrams on the same figure, so that we can see the difference between brick alone, bentonite alone and the composite

3- Response: Thanks very much for this comment. FTIR of bentonite and sawdust were addressed before, and thus, we focused on adding the FTIR spectrum of the prepared BE/SD–MNPs. We request the kind support of Editor and Reviewer to report our results in their current form. 

  1. Page 10-11section « MB adsorption kinetics », the authors must mention at which initial concentration they found these values of kinetic constant, in addition the adsorption kinetics is followed at several initial concentrations of the pollutant
  2. Response: Thanks very much for this comment. The experimental conditions associated to MB adsorption kinetics were reported in section 2.4. Kinetics studies of MB adsorption onto BE/SD–MNPs were considered as follows: A mixture of 25 mg of BE/SD–MNPs and 25 mL of MB solution with a concentration of 150 mg/L was prepared. The adsorbate–adsorbent mixtures were mixed at 220 rpm at different interactions times (i.e. 5, 15, 30, 45, 60, 75, 90, 180 and 360 min) and solution pH 8.0.

  1. Fig 05, authors must add the operating conditions such as: temperature, s/l ratio, time, pH, etc.
  2. Response: Thanks very much for this comment. The operating conditions were added to Figure 5 . Operating conditions: 25 mg of BE/SD–MNPs mass, 25 mL of MB solution (150 mg/L), 25 ºC of adsorption temperature, and solution pH 8.0, please see the corrected article

  1. The authors should determine the specific surface area by BET of their materials.
  2. Response: Actually, some techniques are not available in the institutes and universities of the research group, thus including the BET surface area measurement. To carry out this analysis, more than 60 days can be required. So, we request the Editor and reviewer to kindly accept our article without this information.

  1. The quality of all figures need to be improved
  2. 7. Response: Thanks very much for this comment. The quality of all figures were improved, please see the corrected article

  1. Why has the isoelectric pH not been determined, since it allows to set the ph of the work ?
  2. Response: Thanks very much for this comment. In this study, we determined the surface charge of the BE/SD–MNPs via zeta potential technique and the fabricated composite presented a negative zeta potential of −28.3 mV (Fig. 4c). The ZP value becomes more negative with the increment of solution pH due to the increase of hydroxyl ions and deprotonation of the BE/SD–MNPs functional groups. The highly negative charge of the BE/SD–MNPs surface reflected its high stability against aggregation due to the high repulsion forces between the adjacent particles [36]. Consequently, the solution pH 8.0 is selected to carry out all the MB adsorption experiments.
  3. All Figures: standard deviations should be given.
  4. Response. Thanks very much for this comment. Adsorption isotherms exhibited error bars as given in Figure 6. Overall, all MB adsorption experiments were repeated, and the mean values of the results have been utilized for data evaluation, with the error being always below ± 5 %. We request the kind support of Editor and Reviewer to report our results in their current form.

Round 2

Reviewer 1 Report

-

Author Response

Comment:

English language and style are fine/minor spell check required

Reply: English language and style are  checked and adjused accordingally

Reviewer 2 Report

Thank you for the response, although the reply of comments were not satisfactory. Few changes need to address before accepting the manuscript.

What is the type of sorption of MB whether physisorption and chemisorption in your case?

The term clays/lignocellulosic use only in abstract but not discussed in results discussion. (bentonite/sawdust is more appropriate term).

The terms Kinetics, equilibrium and thermodynamics are common terminologies in adsorption studies, use in title or as keywords.

Author Response

Reviewer 2

  • What is the type of sorption of MB whether physisorption and chemisorption in your case?
  • Response: The sorption of MB was governed by physical interactions ‘’Overall, the adsorption energies were < 40 kJ/mol, which reflected the involvement of the physical forces (e.g., hydrogen bindings, van der Waals, and electrostatic interactions) in MB dye and BE/SD–MNPs interface’’, please see abstract and section 3.7. Interpretation of MB adsorption on BE/SD–MNPs via adsorption energy the corrected article.

  • The term clays/lignocellulosic use only in abstract but not discussed in results discussion. (bentonite/sawdust is more appropriate term).
  • Response: Overall, this study aimed to understand the MB adsorption mechanism using a magnetic clays/lignocellulosic interface like the utilized BE/SD–MNPs composite as a promising strategy in wastewater remediation.

The terms Kinetics, equilibrium and thermodynamics are common terminologies in adsorption studies, use in title or as keywords.

  • Response: The terminologies kinetics, equilibrium and thermodynamics were added to keywords, please see the corrected article

  • English language and style are fine/minor spell check required
  • Response: English language and style are checked and adjusted accordingly

Reviewer 3 Report

Accepted in present form

Author Response

Comment

I don't feel qualified to judge about the English language and style

Reply:

English language and style are checked and adjusted acoordingally